# Cannabis and Palliative Care Utilization among Non-Terminal Cancer Patients in the Illinois Medical Cannabis Program

James A. Croker III [1,2,*], Julie Bobitt [3], Kanika Arora [1] and Brian Kaskie [1,*]

1 Department of Health Management & Policy, University of Iowa, Iowa City, IA 52242, USA
2 Center for Tobacco Control Research and Education, Cardiovascular Research Institute, University of California San Francisco, San Francisco, CA 94158, USA
3 Center for Dissemination and Implementation Science, Department of Medicine, University of Illinois at Chicago, Chicago, IL 61820, USA
* Correspondence: james.croker@ucsf.edu (J.A.C.III); brian-kaskie@uiowa.edu (B.K.)

**Abstract:** High-quality palliative care has been shown to provide benefits for cancer patients, including greater longevity when initiated earlier in treatment. Previous research conducted among terminal patients in the Illinois Medical Cannabis Program has suggested that cannabis may be used as a complement to palliative care and as an alternative to prescription opioid medications. However, there is little research exploring this phenomenon among non-terminal cancer patients receiving palliative care. In this study, we used primary cross-sectional survey data to (1) identify the factors associated with the utilization of palliative care, (2) examine the associations between the utilization of palliative care and self-reported improvements in physical and psychological symptoms, and (3) estimate the differences in the average 30-day pain levels for non-terminal cancer patients receiving palliative care who reported using opioids compared with other non-terminal cancer patients in palliative care who did not report using opioids. In our sample, 87 out of 542 (16%) non-terminal cancer patients were receiving palliative care, and of these 87 patients, 54 (62%) reported opioid use in the past 12 months. Non-terminal cancer patients in the sample who reported low psychological well-being, frequent gastrointestinal symptoms, and prescription opioid use in the past 12 months had greater odds of palliative care utilization. Palliative care utilization was also positively associated with self-reported improvements in gastrointestinal symptoms. The concurrent use of cannabis and prescription opioids was associated with higher average 30-day pain levels and with higher average pain levels at the initiation of cannabis use among those non-terminal cancer patients engaged in palliative care services.

**Keywords:** cancer; palliative care; opioids; medical cannabis

## 1. Introduction

Palliative care offers comprehensive, coordinated, and team-based support for patients with serious or life-threatening conditions [1–4]. Palliative care aims to respond to the patient in a holistic way and rapidly intervene to treat the symptoms and side effects of their disease and treatment while also addressing the related social, psychological, and spiritual problems they may experience. Patients receive palliative care in a variety of settings, including hospitals, outpatient clinics, long-term care facilities, and at home under the direction of a licensed healthcare provider [5]. Palliative care partly emerged from efforts to better respond to the needs of cancer patients and their families in the most advanced stage of disease [6]. Cancer is well situated for palliative care because of the relatively clear and predictable disease trajectory, though diagnosed cancer patients may choose to access palliative care services at any point in their course of care [5,7]. Cancer patients routinely experience severe symptoms resulting from both their condition and treatments, and adequate symptom management is essential for successful treatment outcomes [8–11]. Palliative care for patients with a cancer diagnosis includes many of the same medications

and therapies used to treat their cancer, but with the specific goal of symptom management to reduce suffering and promote patient comfort [5]. Palliative care has routinely been associated with the effective management of the symptoms associated with cancer, along with reductions in adverse effects from chemotherapy and polypharmacy, psychosocial stress to patients and their families, and complexity arising from multimorbidity [12–14]. Moreover, the effective management of symptoms for cancer patients is associated with improved quality of life and greater treatment compliance [15–20]. All cancer patients are eligible to receive palliative care services regardless of age, disease stage, or type of cancer [5]. For some cancer patients, earlier engagement in palliative care services has been associated with increased longevity [21–23]. Because of these potential improvements to the overall quality of care, cancer patients continue to have the most direct access to supportive palliative care services prior to receiving a terminal diagnosis, with clinical recommendations calling for the engagement of these services as an "upstream" intervention, available as close to diagnosis as possible [24,25].

Evidence indicates cannabis use has the potential to enhance palliative care services (excluding *palliative chemotherapy*—that is, chemotherapy given without curative intent but simply to decrease tumor load and increase life expectancy) for non-terminal cancer patients specifically [26–29]. This includes the management of chronic pain, chemotherapy-induced nausea, cachexia, sleep problems, and mental health challenges [30–35]. Chronic pain is likely the most commonly treated physical symptom in palliative care [36]. The management of pain in palliative care is usually accomplished through the administration of prescription opioid medications. Prescription opioids, such as oxycodone and fentanyl, are a class of alkaloid compounds that act on the central nervous system by binding to specific receptors in the brain and spinal cord, resulting in analgesia [37,38]. Prescription opioids are typically used for pain relief, but they can also produce a range of effects, including euphoria and sedation, and potential adverse effects, including various gastrointestinal problems and respiratory depression [39]. Medical cannabis, on the other hand, contains various terpenes and cannabinoids, such as THC and CBD, that interact with the endocannabinoid system in the body [40]. These compounds have been found to have some analgesic, anti-inflammatory, and neuroprotective properties, without the documented risks of chemical dependence that come with the use of prescription opioid medications [41]. When used together in limited amounts, cannabis has been observed to enhance the analgesic effects of prescription opioids [42]. The incorporation of cannabis as a complement in palliative care might also reduce the potential adverse effects from high-dose, long-term use of prescription opioids, including gastrointestinal symptoms such as opioid-induced constipation (OIC), nausea, and vomiting, along with the reduced risk of potential dependence and accidental overdose [43–46]. Figure 1 presents the supportive care continuum for cancer patients considering medical cannabis program enrollment, showing the potential overlap of palliative care and complementary cannabis use at any point in the patient's course of treatment [47–49]. Here, we investigate how patients can choose to receive either curative therapy alone or include palliative care services as part of their treatment at any point after diagnosis. Once a patient receives a terminal diagnosis from their physician, however, the patient then has the option to forgo curative therapy and transition to solely receiving palliative services through hospice care [48].

Within the Illinois Medical Cannabis Program (IMCP), patients are also eligible for cannabis program enrollment immediately upon diagnosis with a qualifying condition, including cancer, allowing them to benefit from this overlap in the supportive care model. Previous research on terminally diagnosed patients in the IMCP suggested that terminal cancer patients receiving hospice or palliative care services were more likely to experience gastrointestinal symptoms, such as OIC, nausea, and vomiting, and more likely to use cannabis to treat them [48,49]. However, while terminal patients receiving hospice or palliative care services were observed to be using cannabis as a *complement* to their care, most terminal patients in the sample, including many cancer patients, indicated they were using cannabis as an *alternative* to palliative care services. Moreover, these patients were

also largely averse to responding to specific survey questions on the topics necessary to pursue palliative care services with their care team and engage those services in a timely fashion.

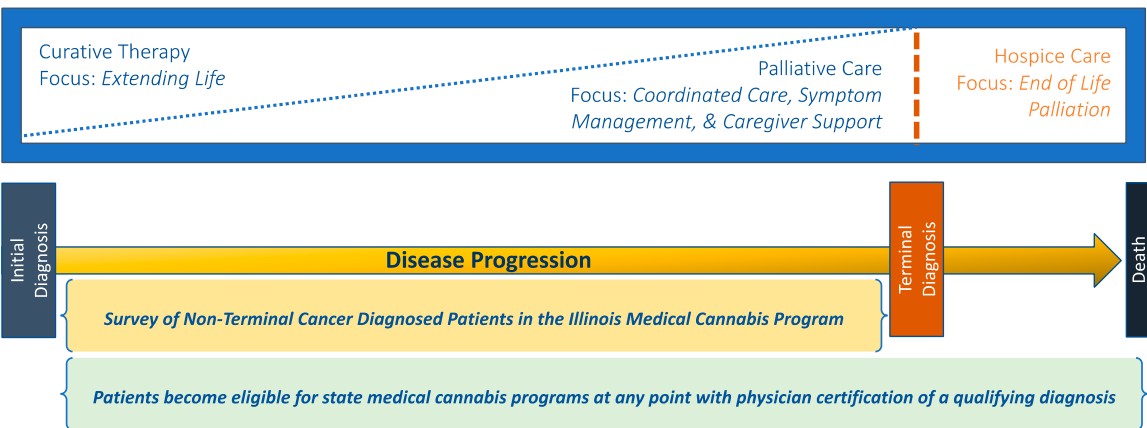

**Figure 1.** Supportive care continuum for cancer patients considering medical cannabis program enrollment.

This study aimed to explore the differences among non-terminal-cancer-diagnosed patients in the IMCP based on their use of palliative care, identify the associations between palliative care utilization and self-reported improvements in a range of health status measures, and identify the differences in pain symptom severity at cannabis use initiation for non-terminal cancer patients receiving palliative care who are also using prescription opioids compared with non-terminal cancer patients receiving palliative care who are not using opioids. We developed three hypotheses to guide our inquiry based on previous observations among IMCP patients with a terminal diagnosis. First, non-terminal cancer patients receiving palliative care would have higher rates of co-diagnosis with musculoskeletal conditions and medical complexity and would engage in a "treatment-focused" approach to cannabis use, i.e., primarily using it for medical purposes, as a complement to prescription opioids, as first-time users later in life, and at their physician's suggestion. Second, palliative care utilization would be associated with positive self-reported improvements in physical symptoms, such as gastrointestinal problems and pain, and psychological symptoms, such as emotional problems, psychological well-being, and health-related quality of life. Third, non-terminal cancer patients receiving palliative care and using prescription opioids would report lower average 30-day pain levels and higher pain intensity levels at cannabis use initiation than those non-terminal cancer patients receiving palliative care but not using opioids. The present study is one of the first to engage non-terminal cancer patients who report using cannabis as a part of their palliative care. We expect that many of the findings observed among terminal patients regarding the adjunctive benefits of palliative care to symptom management will also be seen in non-terminal-cancer-diagnosed patients [48,49].

## 2. Materials and Methods

This study analyzed the primary, cross-sectional data from a survey of medical cannabis users in Illinois to explore the utilization of palliative care by non-terminal cancer patients in the IMCP, examine the differences in self-reported health status for non-terminal-cancer-diagnosed patients receiving palliative care compared with those not receiving palliative care, and detail how the concurrent use of cannabis and prescription opioids is associated with pain symptoms. The survey was an anonymous, closed-access e-mail survey of adults who were enrolled in the IMCP, and the data collection process is described in previous publications [48–52]. Participants were informed of the research purpose, completion time, data storage protocols, and contact information for the study

personnel and provided consent prior to completing the survey. Approval for this research was granted by the Internal Review Board at the University of Illinois at Urbana Champaign (IRB#19223—10/31/2018; transferred to the University of Illinois, Chicago, in 2020, via protocol #2020-1227).

### 2.1. Data

The survey included questions related to health status, cannabis use attitudes and behaviors, experiences with palliative care, opioid use behaviors, and sociodemographic characteristics. Specific questions regarding palliative care utilization, attitudes and experiences with cannabis as a complement, and other medication use were also included. The full survey instrument is available in Supplementary Materials (Supplementary Data S1).

### 2.2. Sample

The sample for the current study included cancer-diagnosed patients participating in the IMPC who report not having received a terminal illness diagnosis from their physician. Terminal illness status was assessed in the survey in multiple ways. First, respondents could self-select a terminal diagnosis as one of their certifying conditions for the program (Q14). Respondents were not prevented from selecting multiple conditions. The survey later asks respondents to self-identify if they are participants in the Illinois Terminal Illness Program (Q17), a fast-track pathway into the IMCP that requires physicians to certify a terminal diagnosis as the qualifying condition for the patient's application. The survey also asks respondents if they had previously received a terminal diagnosis at any point from their physician later in the survey along with the measures of palliative care utilization (Q40a). We considered any report of a prior terminal diagnosis or participation in the Terminal Illness Program as constituting a terminal diagnosis and restricted the data to exclude them. A total of 727 responses from patients who reported receiving a terminal diagnosis from their physician were excluded from the analyses. The data from non-terminal respondents ($n = 3339$) were further restricted to include only those respondents who reported cancer as their qualifying condition for IMCP certification ($n = 542$). The analytic sample structure for non-terminal-cancer-diagnosed patients is shown in Figure 2.

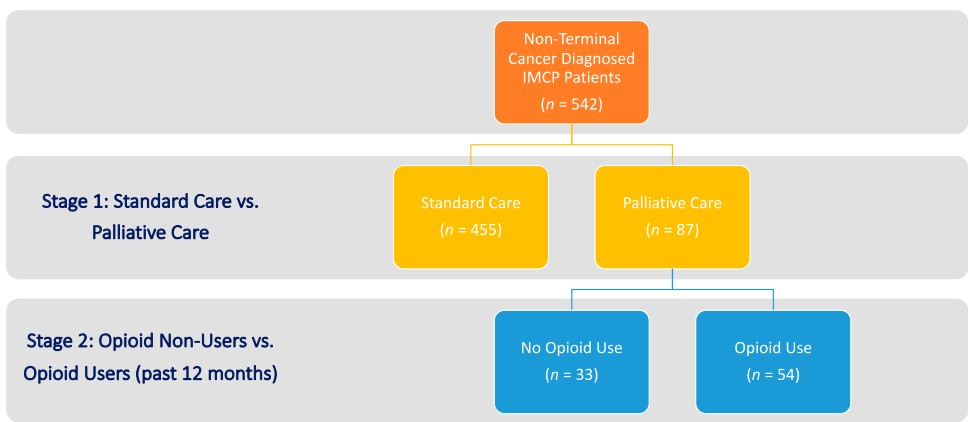

**Figure 2.** Analytic sample structure of non-terminal cancer patients in the IMCP ($n = 542$).

### 2.3. Independent Variables

Demographic variables included in the analyses were age, sex, race/ethnicity, marital status, educational attainment, prior military service, and financial security status. In addition to a self-identified cancer diagnosis qualifying them for the IMCP, the secondary health conditions reported by patients in the program were grouped into categories for analysis, including mental health disorders, musculoskeletal disorders, neurological disorders, and medically complex cases with multiple chronic conditions. Global measures for capturing self-reported physical and mental health status included the ability to manage health status

and psychological well-being. Self-reported symptoms treated with cannabis included gastrointestinal symptoms, pain, and emotional problems.

Cannabis use was assessed with measures capturing purpose (medical only and combined medical and recreational), frequency of use (0–30 days), dosing/administration methods (smoke inhalation, vaporizer, edible products, oral pill/tablet, and cream/ointment), status as a new or naïve cannabis user in later life, and reports of negative experiences with cannabis use in the past year. To identify the potential barriers to accessing the cannabis program, we contrasted the outcomes in terms of the source of patient knowledge about the IMCP, and whether or not the patient's health insurance covered the cannabis certification visit with their doctor. Opioid use was assessed with an indicator for opioid use in the past 12 months [53].

A dichotomous indicator was also included for caregiver proxy response to the survey. Illinois allows cannabis patients to certify a caregiver who is able to purchase and possess cannabis on behalf of the patient. Caregiver proxies were therefore identified given the possibility they may have been the contact email address on file with the IDPH at the time of the survey. When assessing quality and satisfaction in supportive care using questionnaires, proxies have been observed to have higher quality/satisfaction scores while also having more negative reports on clinical nursing and on care coordination [54–58]. This research shows the difference is small but statistically significant. Caregiver proxy response was originally included when the guiding conceptual model was initially developed because of the unique role caregivers play in the treatment process and the potential influence caregivers have on the patient's decision-making regarding care utilization. Informal caregivers have been shown to influence care decisions for non-terminal cancer patients [59]. Because some proxy responses appeared among our sample of non-terminal cancer patients, the indicator was included in the study analyses.

*2.4. Outcome Measures*

In addition to palliative care utilization, the outcomes for this analysis focus on self-reported improvements in health status for gastrointestinal symptoms, pain, emotional problems, psychological well-being, and quality of life. Patients were asked "How does cannabis affect your [health outcome measure]", with three categorical response options ("makes it worse"; "no change"; and "makes it better"). Patients who indicated cannabis use had a positive or negative effect on their general health or specific symptoms were invited to provide a continuous impact rating score via a drag-bar scale (0–100). The outcome measures for the opioid analysis include average pain levels over the past 30 days and at the initiation of cannabis use among non-terminal cancer patients in palliative care. Pain intensity levels were assessed using an 11-point pain scale (0–10, where 0 = "no pain", 1–3 = "mild pain", 4–6 = "moderate pain", 7–9 = "severe pain", and 10 = "worst possible pain").

*2.5. Statistical Analyses*

We began by describing the data and comparing the values for items from our study's guiding conceptual framework using independent *t*-tests for continuous measures and chi-square tests for dichotomous measures to determine statistically significant differences between non-terminal-cancer-diagnosed patients receiving palliative care and non-terminal-cancer-diagnosed patients who were not receiving palliative care. Statistically significant items were identified for the next stage of analysis.

Then, to examine the significant correlates of palliative care utilization among non-terminal cancer patients in the IMCP, we used a logistic regression model to compare non-terminal-cancer-diagnosed patients receiving palliative care with those not receiving palliative care services. Independent variables in the models included categorial age groups (1 = under age 65 years; 2 = age 65–69 years; 3 = age 70–79 years; and 4 = age 80 years and older), sex (male/female), education (less than college/college degree or more), marital status (not married/married), prior military service (non-veteran/veteran), and financial

security status (insecure/secure). Special attention was given to those items identified as statistically significant in the previous step.

To explore the differences in health outcomes, we used a series of ordinary least-square (OLS) regression models to identify the associations between self-reported improvement scores (0–100) and outcomes (gastrointestinal symptoms, pain, emotional problems, general psychological well-being, and health-related quality of life), among respondents who indicated cannabis was used to treat it. The independent variables in the OLS models included indicators for palliative care utilization, frequency of cannabis use (0–30 days), opioid use in the past year, and medical complexity, along with the indicators for demographics and caregiver proxy. The linear regressions took the following general form:

$$\Upsilon_i = \beta_{0i} + \beta_{1i}\,pc + \beta_{2i}\,frequency + \beta_{3i}\,opioids + \beta_{4i}\,demos + \beta_{5i}\,proxy + \beta_{6i}\,complex + \varepsilon_i$$

where $\Upsilon$ represents the improvement score (0–100) for the outcome measure (i.e., gastrointestinal symptoms, pain, emotional problems, general psychological well-being, and health-related quality of life), "pc" is an indicator for palliative care utilization, "frequency" captures 30-day past cannabis use (0–30 days), "opioids" is an indicator for the concurrent use of cannabis with prescription opioids, "demos" is an indicator for the demographic measures (age group category, sex, race, education, marital status, and financial status), "proxy" is an indicator for surveys completed by caregiver proxy, and "complex" captures medical complexity associated with their diagnosed conditions.

Finally, to examine the associations between the concurrent use of cannabis and prescription opioids and pain management among non-terminal-cancer-diagnosed patients receiving palliative care, we used independent *t*-tests to assess the differences in average pain levels over the past 30 days, and at the initiation of cannabis dosing.

Response rates for some measures were reduced as a function of the adaptive questionnaire. Given that we performed multiple statistical tests, the Benjamini–Hochberg procedure (BHP) for controlling the false-positive rate from multiple comparisons was used [60]. BHP-adjusted *p*-values are presented in the tables. A BHP-adjusted *p*-value of 0.05 or less was considered statistically significant on all two-tailed tests. All data management and statistical analyses were performed using Stata (16.1, StataCorp LLC, College Station, TX, USA). The report here is consistent with the CHERRIES and STROBE checklists for cross-sectional studies [61,62].

### 3. Results

Of the 542 non-terminal-cancer-diagnosed patients in the IMCP, 87 (16%) reported receiving palliative care. Table 1 presents the measures of central tendency and the results of the tests of item significance by palliative care utilization status. Table 2 presents the results of the logistic regression comparing non-terminal cancer patients receiving palliative care services to those not using palliative care. When controlling for demographic variables and items pulled from the previous stage of analysis, we found significant differences in health status between non-terminal cancer patients receiving palliative care and those who did not. We found increased odds of palliation for non-terminal cancer patients with low psychological well-being (2.28 (0.93), $p < 0.05$), frequent gastrointestinal symptoms (1.86 (0.57), $p < 0.05$), and self-reported prescription opioid use in the past 12 months (1.77 (0.45), $p < 0.05$). However, we found lower odds of palliative care utilization for females (0.56 (0.16), $p < 0.05$) compared with males, married persons (0.58 (0.15), $p < 0.05$) compared with unmarried persons, and for those who identified as financially secure (0.55 (0.15), $p < 0.05$) compared with those who experienced financial insecurity.

**Table 1.** Descriptive statistics and tests of significance by palliative care utilization status (*n* = 542).

| Characteristics [†] | Non-PC [a] | PC | *p*-Value * |
| --- | --- | --- | --- |
| | Perc. [b] (SE) [c] (*n* = 455) | Perc. (SE) (*n* = 87) | |
| Demographics | | | |
| Age in years (Mean (SE)) | 67.37 (0.26) | 67.15 (0.63) | 0.65 |
| Under 60 years old | 0.01 (0.01) | 0.01 (0.01) | 0.41 |
| Age 60–64 years | 0.37 (0.02) | 0.40 (0.05) | 0.49 |
| Age 65–69 years | 0.33 (0.02) | 0.31 (0.05) | 0.69 |
| Age 70–79 years | 0.28 (0.02) | 0.25 (0.05) | 0.59 |
| Age 80 years and older | 0.02 (0.01) | 0.04 (0.02) | 0.86 |
| Female | 0.43 (0.02) | 0.38 (0.05) | 0.19 |
| Non-white | 0.11 (0.02) | 0.18 (0.04) | 0.10 |
| College degree or more | 0.49 (0.02) | 0.44 (0.05) | 0.51 |
| Married | 0.69 (0.02) | 0.62 (0.05) | 0.19 |
| Military veteran | 0.18 (0.02) | 0.20 (0.04) | 0.34 |
| Currently employed | 0.24 (0.02) | 0.19 (0.04) | 0.25 |
| Financially secure | 0.76 (0.02) | 0.67 (0.05) | 0.08 |
| Health status | | | |
| Caregiver proxy | 0.05 (0.01) | 0.06 (0.03) | 0.72 |
| Physically disabled | 0.22 (0.02) | 0.30 (0.05) | 0.08 |
| Difficulty managing health status | 0.14 (0.02) | 0.20 (0.04) | 0.21 |
| Low psychological well-being | 0.15 (0.02) | 0.28 (0.05) | <0.001 |
| Average 30-day pain level (0–10) | 4.47 (0.13) | 4.67 (0.26) | 0.46 |
| Frequent emotional problems | 0.21 (0.02) | 0.23 (0.05) | 0.49 |
| Frequent gastrointestinal symptoms | 0.22 (0.02) | 0.35 (0.05) | 0.01 |
| Diagnosed neurological disorder | 0.10 (0.02) | 0.06 (0.03) | 0.35 |
| Diagnosed mental health condition | 0.08 (0.01) | 0.09 (0.03) | 0.74 |
| Diagnosed musculoskeletal disorder | 0.21 (0.02) | 0.17 (0.04) | 0.58 |
| Medically complex | 0.01 (0.01) | 0.01 (0.01) | 0.81 |
| Using cannabis to treat pain | 0.77 (0.02) | 0.84 (0.04) | 0.12 |
| Using cannabis to treat emotional problems | 0.39 (0.03) | 0.46 (0.06) | 0.07 |
| Using cannabis to treat gastrointestinal symptoms | 0.39 (0.03) | 0.54 (0.06) | 0.01 |
| Prescription opioid use in the past year | 0.38 (0.02) | 0.61 (0.05) | <0.001 |
| Cannabis Use and Program Access | | | |
| Using cannabis for medical purposes only | 0.66 (0.02) | 0.70 (0.05) | 0.37 |
| Using cannabis for combined medical and recreational purposes | 0.33 (0.02) | 0.29 (0.05) | 0.40 |
| Frequency of cannabis use (0–30 days) (Mean (SE)) | 20.75 (0.49) | 22.22 (1.15) | 0.24 |
| Cannabis dosing via smoke inhalation | 0.47 (0.02) | 0.52 (0.05) | 0.30 |
| Cannabis dosing via oral pill/tablet | 0.20 (0.02) | 0.21 (0.04) | 0.96 |
| Cannabis dosing via edible products | 0.63 (0.02) | 0.52 (0.05) | 0.09 |
| Naïve cannabis user | 0.28 (0.02) | 0.31 (0.05) | 0.67 |
| Knowledge of cannabis program from physician | 0.39 (0.02) | 0.43 (0.05) | 0.55 |
| Insurance coverage of certification visit | 0.42 (0.02) | 0.48 (0.05) | 0.32 |
| Negative cannabis use experience in the past year | 0.11 (0.02) | 0.08 (0.03) | 0.38 |

[†] For continuous variables, independent *t*-tests were used for tests of statistical associations; for dichotomous measures, chi-square tests were used. * Benjamini–Hochberg procedure (BHP)-adjusted *p*-values are presented. [a] PC—Palliative Care; [b] Perc—Percent; [c] SE—Standard Error.

**Table 2.** Logistic regression[†] predicting palliative care utilization: comparing non-terminal-cancer-diagnosed patients on odds of palliative care use (*n* = 487).

| Palliative Care Patients | AOR [a] | [95% CI] | *p*-Value * |
|---|---|---|---|
| Demographics | | | |
| Female | 0.56 | [0.33–0.97] | 0.04 |
| Married | 0.58 | [0.35–0.96] | 0.03 |
| Financially secure | 0.55 | [0.32–0.92] | 0.02 |
| Health status | | | |
| Low psychological well-being | 2.28 | [1.03–5.06] | 0.04 |
| Frequent gastrointestinal symptoms | 1.86 | [1.03–3.38] | 0.04 |
| Opioid use in the past year | 1.77 | [1.07–2.90] | 0.03 |

[†] Logistic regression included indicators for age group category, race/ethnicity, education, veteran status, caregiver proxy use, physical disability, difficulty managing health status, frequent emotional problems, musculoskeletal conditions, mental health conditions, treating gastrointestinal symptoms, use of cannabis as an opioid complement, days using cannabis, naïve cannabis user, and negative cannabis use experience in the past year as covariates. Total *n* for this analysis is slightly reduced due to missing data. * Benjamini–Hochberg procedure (BHP)-adjusted *p*-values are presented. [a] AOR= adjusted odds ratio.

### 3.1. Self-Reported Outcome Improvements

When carrying out the OLS regression modeling of the measures capturing symptom changes from cannabis use, we found that palliative care utilization was significantly associated with self-reported improvements in gastrointestinal symptoms ($\beta$(SE) = 12.44 (4.67), $p < 0.01$). The frequency of cannabis use in days (0–30 days) had consistent positive associations across the range of outcomes, with coefficients ranging from 1.51 (0.162) to 1.65 (0.110) ($p < 0.001$). The concurrent use of prescription opioids in the past year had its largest associations with psychological well-being ($\beta$(SE) = 11.32 (2.962), $p < 0.001$) and pain ($\beta$(SE) = 7.81 (2.919), $p < 0.01$). The regression coefficients for the instrumental variables are shown in Table 3.

**Table 3.** OLS [†] regression coefficients of self-reported improvements to health outcomes.

| Instrumental Variables | GI [a] (*n* = 193) Coef. (SE) | Pain (*n* = 347) Coef. (SE) | EMO [b] (*n* = 241) Coef. (SE) | PSY [c] (*n* = 278) Coef. (SE) | QOL [d] (*n* = 380) Coef. (SE) |
|---|---|---|---|---|---|
| Palliative care service utilization | 12.44 ** (4.67) | 6.18 (3.80) | 7.54 (4.17) | 3.66 (3.62) | 3.02 (3.40) |
| Frequency of Cannabis use (0–30 days in past 30 days) | 1.51 *** (0.16) | 1.58 *** (0.12) | 1.59 *** (0.14) | 1.64 *** (0.13) | 1.65 *** (0.11) |
| Opioid use in the past year | 6.84 (3.77) | 7.81 ** (2.92) | 8.06 * (3.26) | 11.32 *** (2.96) | 6.65 * (2.65) |
| Medically complex classification | 19.61 (26.73) | 4.28 (15.83) | 11.52 (17.64) | 7.40 (13.85) | 2.71 (12.78) |

[†] Ordinary least-square regressions included demographic covariates (i.e., age group category, sex, race/ethnicity, marital status, educational attainment, prior military service, and financial security status) along with an indicator for caregiver proxy. [a] = gastrointestinal symptoms; [b] = emotional problems; [c] = psychological well-being; [d] = quality of life. * Benjamini–Hochberg procedure (BHP)-adjusted *p*-values presented, * = $p < 0.05$; ** = $p < 0.01$; *** = $p < 0.001$.

### 3.2. Opioid Use among Non-Terminal Cancer Patients Receiving Palliative Care

In total, 224 of the 542 (41%) non-terminal-cancer-diagnosed patients in our sample reported using prescription opioids in the past year. Among the non-terminal-cancer-diagnosed patients in palliative care, 54 of 87 (62%) patients reported opioid use in the past 12 months, and 49 of the 54 (91%) patients in palliative care who were also using opioids indicated their cannabis use was intended as a complement. Table 4 presents the results of independent *t*-tests examining the differences in average pain levels for palliative

care patients relative to opioid use behavior in the past 12 months. The concurrent use of cannabis and prescription opioids was associated with higher average 30-day pain levels (mean (SE) = 1.82 (0.491), *p* < 0.001) and higher pain levels at initiation of cannabis use (mean (SE) = 2.18 (0.577), *p* < 0.001) among non-terminal cancer patients receiving palliative care.

**Table 4.** Independent *t*-tests of differences in mean pain levels among non-terminal cancer patients receiving palliative care relative to opioid use.

| Self-Reported Average Pain Levels | Opioid Non-Users (*n* = 33) Mean (SE) | Opioid Users (*n* = 54) Mean (SE) | Difference Mean (SE) | *p*-Value * |
|---|---|---|---|---|
| Pain level[†] at initiation of cannabis dosing | 3.89 (0.53) | 6.07 (0.23) | 2.18 (0.58) | <0.001 |
| Average 30-day pain level [†] | 3.56 (0.38) | 5.38 (0.32) | 1.82 (0.49) | <0.001 |

[†] The scores are 0–10, where 0 = "no pain", 1–3 = "mild pain", 4–6 = "moderate pain", 7–9 = "severe pain", 10 = "worst possible pain". * BHP-adjusted *p*-values are presented.

## 4. Discussion

This study sheds light on the intersection between palliative care utilization and cannabis use among non-terminal cancer patients by analyzing the cross-sectional survey data collected from participants in the IMCP. Our findings show that the non-terminal cancer patients in our sample who were receiving palliative care were statistically different in their health status from those patients not receiving palliative care, specifically in terms of reporting gastrointestinal issues and cannabis use behaviors, particularly how they intend to use cannabis and manage their use as a therapeutic and a complement to prescription opioids.

The first hypothesis of the study is partially supported, indicating significant differences in health status between palliative care and non-palliative care patients in terms of psychological well-being, treatment of gastrointestinal symptoms, and opioid use in the past year. The findings did not support increased odds of palliative care utilization for individuals who qualify as medically complex. The lower odds of palliative care utilization observed among females, married individuals, and those not in financial distress in this study are consistent with previous research on terminal patients from the IMCP [47,48]. However, these findings are not entirely consistent with findings from a larger study of cancer patients in the New York program that indicated cancer patients using cannabis were more likely to be female [63]. Previous research has shown that cancer patients in particular have strong attitudes in favor of the legalization of cannabis use for therapeutic/medical purposes and believe in its benefits for treating cancer-related symptoms [64]. We found solid evidence in support of the hypothesis that non-terminal cancer patients receiving palliative care pursued a treatment-focused approach where cannabis was primarily used for medical purposes, and complementary cannabis was used to enhance the effect of prescription opioids; we also found relatively high rates of respondents who were "naïve" users and those who reported using cannabis specifically at the suggestion of their physician. Previous studies have shown mixed attitudes among physicians in terms of their willingness to recommend cannabis to patients as a treatment option and generally high levels of concern regarding its use, particularly among US physicians [65,66]. However, some evidence suggests greater acceptance of applications of medical cannabis specifically for cancer-diagnosed patients [67].

The second hypothesis is relatively supported by evidence indicating that palliative care utilization has positive associations with self-reported improvements in physical symptoms. While we had hoped to see significant associations between palliative care utilization and the full range of self-reported health status measures, some evidence of effective physical symptom management with palliation was observed in the linear associations between palliative care and the measures capturing self-reported improvement in gastrointestinal

symptoms and pain levels for those patients receiving palliative care who were also using prescription opioids. However, no statistically significant associations were observed for palliative care utilization in measures related to psychological symptoms. We were surprised to see that the largest, statistically significant improvements in psychological well-being were associated with prescription opioid use in the past 12 months. However, it seems likely that the effective management of pain via prescription opioids in palliative care would correspond to the availability of high-quality psychological and spiritual support as part of a comprehensive, coordinated palliative care service model [68].

Finally, the third hypothesis is supported by the data indicating lower average 30-day pain levels and higher pain levels at cannabis use initiation for non-terminal cancer patients receiving palliative care who also reported using opioids compared with those non-terminal cancer patients in palliative care who were not using prescription opioids. This study suggests that patients using cannabis as a complement to opioids may have generally well-managed pain symptoms via prescription medications but may primarily use cannabis for the so-called "breakthrough" pain, that is, brief episodes when prescription opioid medications are insufficient for analgesia [69]. If this is the case, future clinical studies should specifically look at pain level reports with measures for current opioid use at varying dosages and should seek to determine the appropriate means and quantity of cannabis to be used as a complement to prescription opioid medications in such contexts. Doing so could allow for greater insight into the use of cannabis as a complementary therapy in palliative care for particular kinds of cancers and for managing specific physical symptoms. While we are hesitant to recommend the application of cannabis as a complement to palliative care for any specific type of cancer, this research suggests that patients who report frequent gastrointestinal issues and breakthrough pain episodes may be in the best position to benefit from the application of cannabis as part of palliative care.

*Limitations*

Although this study provides valuable findings, some limitations should be taken into account. Firstly, the cross-sectional design of the study does not allow for the determination of any causal relationships. Moreover, self-reported data may be subject to recall bias or under-/over-reporting, particularly in regard to the patient's self-reported disease state and prior terminal diagnosis. The use of state-level data and the variability in medical cannabis regulations across states may also limit the generalizability of the findings. Additionally, the limited sample sizes of patients receiving palliative care and using prescription opioids may also be limitations. Nonetheless, the findings of this study provide insights for future hypothesis development and testing, including specific research questions and potential biological measures for examination in clinical studies. A specific next step for research would be a longitudinal panel study of patients with varying cancer diagnoses to test for long-term positive associations, symptom management with medical cannabis, and differences in self-reported pain levels at varying dosing levels of the combined use of cannabis and opioid medication.

## 5. Conclusions

This research offers greater insight into different pathways for symptom management among the non-terminal-cancer-diagnosed patients enrolled in the IMCP. This study adds to the literature by engaging a sample of cannabis-using patients and identifying the role cannabis plays in managing specific symptoms and adverse effects from traditional treatments for this non-terminal cancer population. The findings here suggest cannabis works well as a complementary therapy for non-terminal cancer patients receiving palliative care. However, it also shows that, at least among our sample, cannabis is most frequently used as an alternative means of symptom management, being used instead of supportive palliative care services or prescription opioid medications. Given that both palliative care service utilization and higher cannabis use frequency were associated with improved outcome scores, the question arises as to whether these improvements are meaningfully

different among these populations over time. Future research on outcome effects is needed using larger longitudinal randomized samples.

**Supplementary Materials:** The following supporting information can be downloaded at: https://www.mdpi.com/article/10.3390/psychoactives2010004/s1, Data S1: Full survey instrument.

**Author Contributions:** Conceptualization, J.A.C.III; methodology, J.A.C.III and K.A.; software, J.A.C.III; validation, J.B., K.A. and B.K.; formal analysis, J.A.C.III; investigation, J.A.C.III; resources, J.B. and B.K.; data curation, J.B.; writing—original draft preparation, J.A.C.III; writing—review and editing, J.B., K.A. and B.K.; visualization, J.A.C.III; supervision, B.K.; project administration, J.A.C.III; funding acquisition, J.B. All authors have read and agreed to the published version of the manuscript.

**Funding:** This research was funded by the Illinois Department of Public Health. Agreement No. 93107003G.

**Institutional Review Board Statement:** The study obtained IRB approval from the University of Illinois at Urbana Champaign.

**Informed Consent Statement:** Informed consent was obtained from all individuals prior to completing the survey.

**Data Availability Statement:** The data that support the findings of this study are available from the Illinois Department of Public Health. Restrictions apply to the availability of these data, which were used under license for this study, given the sensitive nature of the patient information contained.

**Acknowledgments:** We would like to thank Karen Mancera-Cuevas, Paula Atteberry, and Elaine Ewing from the Illinois Department of Public Health for their help with project development. We would also like to thank Dan Shane, Sara Sanders, Keith Mueller, Gary Milavetz, and Divya Bhagianadh at the University of Iowa; Hyojung Kang, and Laura Quintero Silva at the University of Illinois Urbana Champaign; and other researchers associated with the Cannabis and Older Persons Study project at the University of Iowa.

**Conflicts of Interest:** The authors declare no conflict of interest.

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
