# Peer review of "Cannabis and Palliative Care Utilization among Non-Terminal Cancer Patients in the Illinois Medical Cannabis Program"

_psychoactives, doi:10.3390/psychoactives2010004_

Round 1

Reviewer 1 Report

This article sheds light on an interesting and important topic, and highlights the need for further investigation into the varied benefits (and harms) cannabis can provide to cancer patients. 

Some items I would like to bring attention to that may need some further clarification and/or minor edits: 

Lines 45-46 - "enhance non-chemotherapy palliative care services" - I am unclear as to what kinds of services this is specifically referencing. 

Lines 62-67 - some recommendations: Instead of "in supportive" say "receiving palliative." Given this is a very long sentence I would put a period  after "services" in line 65 and start the next sentence with "They were..."

Line 255 - "While we had hopeD..."

Line 269 - ...the reality THAT THIS has limited..."

As for the survey that was completed by the patients, it would be helpful to see what the survey questions are without having to look through separate papers. How are the patients being designated as "non-terminal"? Is this self-report? If so, it should be mentioned that this is flawed given the well known inaccuracies of patients knowing their cancer staging and prognosis. 

Discussion as to why it is believed that patients who were using cannabis and opioid concurrently had higher average pain levels. Is it believed that cannabis may be potentiating the pain? That the concurrent usage of these two agents is a result of this cohort being in higher pain overall?

Overall I think this is a nice study that prompts further investigation on this topic. 

Reviewer 2 Report

The work presented by James A Croker III and colleagues provides interesting data regarding the supportive care in non-terminal cancer patients and how to improve the their overall quality of life throughout a Cannabis program. Nonetheless, the manuscript is not easy to read and many points need to be further emphasized. Above all a re-written of some aspects is required.

From my point of view these main points have to be addressed before the manuscript is available for publication in this journal:

1) Introduction: Indicate a) what is a palliative care in cancer patients, b) the different between non-terminal and terminal cancer patients in order to receive or not this supportive care specifying the use of Cannabis and or opioids, c) a general paragraph explaining that opioids are alkaloids, Cannabis comes from anthraquinones and terpenes pathway and their physiological roles etc, d) Sentence of lines 45-46 is not clear, non-chemotherapy is radiotherapy, right? Please explain, e) Which are the objectives proposed regarding Cannabis and opioids combination in this kind of patients? Could you specify some types of specific cancers?

2) Material an Methods: a) minor mistake in sentence 92, b) 2.1 Data: survey should be provided, c) 2.3 and 2.4 there is no references along the text, d) 2.5: line 143: “two groups” Which ones?. Then in the footnotes or tables 1 and 2 it is not detailed which test has been used

3) Results: Table 1 and Table 2 seem to be analyzed in the same way since there is no footnotes to explain how it has been performed and the test used (in Table 2 there is a logistic regression). Table 1 is related to n= 542 and Table 2 (487) but in 2.5 there are two groups (stage 1 or stage 2…). Mean and (SE) should be further explained, for instance Non-white 0,11 what does it mean? 11%? Or how was it measured?

4) Discussion and conclusion: Which is the implication of the higher pain level at initiation of the Cannabis use? How this hyperalgesia could be explained? For which kind of patients opioids + Cannabis should be made? I think this clarification could shed light into the use of Cannabis as a complementary therapy for particular kind of cancers in palliative care managing specific symptoms.

Reviewer 3 Report

The authors performed a retrospective analysis of cross-sectional data from a survey of medical cannabis users in Illinois. The objective was to explore differences among non-terminal cancer-diagnosed patients in the Illinois Medical Cannabis Program (IMCP) regarding their use of palliative care, identify associations between palliative care utilization and self-reported improvements on different health status measures, and identify differences in pain symptom severity at cannabis use initiation for cancer patients in palliative care using opioids compared to those in palliative care not using opioids. The topic is up-to-date and of interest. The manuscript contains a large amount of data, but is difficult to read and often does not describe clearly/thoroughly enough what was done or what sample is being referred to. It shouldn’t be the reader’s responsibility to determine with a lot of effort what the authors have been done here. Additionally, the conclusions made are not supported by the results found and the discussion is not focusing on important findings and isn’t very comprehensive. Overall, in my opinion, the manuscript in its current form has significant shortcomings and needs considerable further work for publication.

Abstract:

-         Line 19: which sample are you referring to here, all the non-terminal cancer patients or the non-terminal cancer patients using palliative care? Please specify.

Introduction:

-         P. 1, line 33: I would remove “curative” and only write treatments

-         P. 1, line 41: what do you mean by “because of this”, what are you referring to? Please specify.

-         Supportive care: please indicate/define what you specifically include herewith. Are you using this term for palliative care in general or just specific aspects? This might also be better specified in the title.

-         P. 2, lines 71-85, objectives/hypotheses: please not only indicate «cancer patients» but specify that you were focusing on non-terminal cancer patients (or indicate if/where you didn’t).

-         P. 2., line 76/77: why are you hypothesizing that cancer patients in palliative care will have significantly higher rates of co-diagnoses of musculoskeletal conditions, on what is this based on?

-         P. 2, lines 75-86: Please describe/rewrite your hypotheses more clearly. At the moment, you mention several aspects or are enumerating several different hypotheses in one sentence, so it’s very difficult to determine what you mean or what you are focusing on.

Figures:

-         Fig. 1: please adapt the figure yourself or also use the description and references in your figure legend. Please also describe all the abbreviations in the figure legend (TIP, EOL). 

-         Fig. 2: I would suggest using «non-terminal» cancer-diagnosed IMCP patients here too

Tables:

-         Table 1: what do these values represent, percentages? Please indicate.

Materials and methods:

-         P. 3, lines 116-118: could you specify what you mean by “ability to manage health status”? how was this assessed, could you provide us with the specific question, or was this even evaluated with a questionnaire?

-         P. 4, line 127: what do you mean by “indicator for opioid use”? Did you assess a general use of opioids, or did you use the frequency and amount of opioid use? Was the opioid use some time during the past year or ongoing? Please specify. It would make a huge difference if you assessed current pain levels and health status if you only did assess past-year opioid use and grouped the respondents according to this and not according to ongoing/current opioid use. This needs clarification.

Design/statistics:

-         P.3, line 93: the authors state, that the design of the survey instrument and the data collection process are described in previous publications. Still, I would like to know specifically for this project, which questions were asked (especially how the cannabis use attitudes and experiences with palliative care were assessed, what was included herein?).

-         P.3, line 105: what exactly is considered a terminal diagnosis? Are there exact conditions to choose from? Or were they just indicating the disease from which they were suffering? On what is this based, life expectancy or treatment options? How do patients themselves know that/if they’re terminal (or not)? Was this objectively validated? This poses a significant concern for me, that respondents were “not reporting a terminal diagnosis” without giving us further information, what this means explicitly and how this was verified. This should be evaluated further.

-         Overall, very small sample size for patients receiving palliative care and additionally subdivided by opioid use, this further limits the value of the analysis – this should at least be noted in the limitations.

-         Statistical analysis section (overall): please clearly indicate which groups you are comparing for the specific analysis, this is not clear to me. For example, p. 4., line 143: between the two groups, presumably PC vs. Non-PC based on results, but not clear from reading this section for the first time without the following results. This should be clarified and made easily recognizable and comprehensible throughout the section.

-         P. 4, line 144: n=542: this does represent the whole sample? Why is this n mentioned in combination with significant items that were pulled to the next stage of analysis? I assume these weren’t 542 items, please clarify.

-         Please clarify what you mean by “key items” (p. 4, line 142).

-         P. 4, line 146: why did you assess significant correlates of palliative care utilization among terminal IMCP patients? Is this incorrectly written? According to previous descriptions, I assumed that you evaluated non-terminal IMCP patients.

-         P. 4, line 161: what is meant by caregiver proxy? This has never been mentioned before, what did you assess here? Later (line 169), it is mentioned that these correspond to surveys completed by caregiver proxies, what is the rationale for including this as an independent variable in the OLS model? Please indicate why you included this in the analysis.

-         Why did you include so many variables as covariates in the logistic regression, what was the rationale (you mentioned that you controlled for demographic variables and items pulled from the previous stage of analysis, but not all of these items did significantly differ between groups before)? Please elaborate further on how you exactly performed this analysis and why you controlled for all these items.

Results:

-         P. 5, lines 190-193: not all values mentioned here match the values reported in Table 2, please clarify and indicate specifically which values are reported here and the significance of asterisks.

-         P. 7, line 207/208: association reported for palliative care utilization with the self-reported improvement of GI symptoms doesn’t match with Table 3, please clarify.

Discussion:

-         Overall weak discussion with primarily repetition and (sometimes wrong) summary of results without proper discussion of the findings or next steps/assessments.

-         P. 8, line 235-236: superficial summary, which is not accurate; e.g., according to your analyses, PC-using non-terminal cancer patients did not statistically differ in demographics and program access and cannabis use in general (only in intention to use for gastrointestinal symptoms). Please clarify this statement and describe the findings more thoroughly.

-         P. 8, line 245: Please reference the previous research with similar findings regarding lower odds of palliative care utilization on demographic measures.

-         P. 8, line 247: you cite a study that found increased likeliness to use cannabis for female cancer patients; in my opinion this doesn’t contradict the finding for less palliative care utilization in female cancer patients also using cannabis.

-         P. 8, line 250-253: Similarly as above, this statement is not entirely accurate: I would not say “exclusively” with 70% PC-using cancer patients using cannabis for medical purposes but rather “primarily”; the rate of naïve users specifically, but also the prevalence of cannabis use at the suggestion of their physician cannot be considered “high” (30 and 43%, respectively). Also, all these findings (except complementary cannabis use to enhance the effect of prescription opioids) did not differ from the non-PC using cancer group, so this statement is misleading as it gives the impression that this is only valid for cancer patients in palliative care (state “cancer patients generally”?). Please clarify this statement.

-         P. 8, lines 255-259: Please further elaborate and discuss the evidence you’ve found in support of your second hypothesis; this paragraph as is doesn’t add any value to the manuscript.

-         Limitations, p. 9, lines 269-270: please rephrase this sentence, it makes no sense as is, probably just misspelled a word.

-         P. 9, lines 271-272: Please indicate what future research questions could be addressed based on your findings, otherwise vacuous sentence.

Round 2

Reviewer 3 Report

I want to thank the authors for the comprehensive review. All my issues have been appropriately addressed, and I think the manuscript quality has significantly increased.